# Unraveling Heterogeneity of Coral Microbiome Assemblages in Tropical and Subtropical Corals in the South China Sea

**DOI:** 10.3390/microorganisms8040604

**Published:** 2020-04-21

**Authors:** Sanqiang Gong, Xuejie Jin, Lijuan Ren, Yehui Tan, Xiaomin Xia

**Affiliations:** 1Key Laboratory of Tropical Marine Bio-resources and Ecology & Guangdong Provincial Key Laboratory of Applied Marine Biology, South China Sea Institute of Oceanology, Chinese Academy of Sciences, Guangzhou 510301, China; gongsanqiang@scsio.ac.cn (S.G.); jinxj@scsio.ac.cn (X.J.); tanyh@scsio.ac.cn (Y.T.); 2Southern Marine Science and Engineering Guangdong Laboratory (Guangzhou), Guangzhou 510301, China; 3Department of Ecology and Institute of Hydrobiology, Jinan University, Guangzhou 510632, China; lijuanren@jnu.edu.cn

**Keywords:** coral, microbiome, holobiont, thermal regimes, South China Sea

## Abstract

Understanding the coral microbiome is critical for predicting the fidelity of coral symbiosis with growing surface seawater temperature (SST). However, how the coral microbiome will respond to increasing SST is still understudied. Here, we compared the coral microbiome assemblages among 73 samples across six typical South China Sea coral species in two thermal regimes. The results revealed that the composition of microbiome varied across both coral species and thermal regimes, except for *Porites lutea.* The tropical coral microbiome displayed stronger heterogeneity and had a more un-compacted ecological network than subtropical coral microbiome. The coral microbiome was more strongly determined by environmental factors than host specificity. γ- (32%) and α-proteobacteria (19%), Bacteroidetes (14%), Firmicutes (14%), Actinobacteria (6%) and Cyanobacteria (2%) dominated the coral microbiome. Additionally, bacteria inferred to play potential roles in host nutrients metabolism, several keystone bacteria detected in human and plant rhizospheric microbiome were retrieved in explored corals. This study not only disentangles how different host taxa and microbiome interact and how such an interaction is affected by thermal regimes, but also identifies previously unrecognized keystone bacteria in corals, and also infers the community structure of coral microbiome will be changed from a compacted to an un-compacted network under elevated SST.

## 1. Introduction

The coral reef ecosystems are the most biologically diverse ecosystems on earth, providing vital ecosystem goods and services in the ocean [1,2]. Despite their importance, coral reefs around the world ocean are declining due to bleaching events and emerging diseases induced by global climate change and anthropogenic disturbances [3]. Previous studies have indicated that the ecological success of coral reefs in oligotrophic seawater majorly relies on symbiosis of the cnidarian coral with their hosted microorganisms, including algae (especially Symbiodiniaceae), bacteria, virus, etc. [4,5,6]. Since 1971, the coral-Symbiodiniaceae symbiosis has been well understood [7], however the revealing of bacteria associated with coral (this study refers to coral microbiome) is rather limited. 

Recent studies using 16S rRNA gene pyrosequencing have revealed that the coral microbiome is one of the most complex microbial biospheres, including thousands of bacterial phylotypes [8,9]. In general, γ- and α-proteobacteria dominate the coral microbiome. Members of the phyla Bacteroidetes, Firmicutes, Actinobacteria and Cyanobacteria are also abundant [9]. The consistency of dominant coral microbiome between individual coral samples has been investigated recently [8,10,11,12,13,14,15]. One of the most important findings of these studies is the substantial variability of coral microbiome among individuals. Furthermore, some researchers revealed that although corals acquire their associated bacteria from surrounding seawater during early life stages, their microbiome composition is distinct from that of the surrounding seawater, suggesting the specific control of the coral microbiome by hosts [10,13]. On the other hand, changes in the coral microbiome response to geographic locations, elevated SST, salinity, nutrients, etc. have also been documented [16,17,18]. Notably, Brener-Raffalli et al. [19] reported that thermal regime drives bacterial assemblages in the *Pocillopora damicornis* living in different geographical locations of the Pacific Ocean. However, possible taxon-specific differences in coral microbiome need further validation, and a comprehensive analysis of the coral microbiome assemblages considering both host taxa and natural thermal regimes needs to be conducted.

The South China Sea (SCS) is an ideal site to study coral microbiome assemblages in different host taxa and thermal regimes. It has extensive coral reef developed (at least 8000 km^2^, including at least 571 coral species) from the south tropical reef regions (e.g., Nansha Islands, Zhongsha Islands, Xisha Islands, Hainan Island, Leizhou Peninsula) to the north subtropical reef regions (e.g., Xiaolajia, Dalajia and Sanmen Islands of Daya Bay) [20,21]. Among these reef regions, the Sanya Bay reef (Hainan Island, China) has a typical tropical ocean climate with a mean annual SST of 27 °C [22]. The Daya Bay reef is located on the southeast of Shenzhen City (Guangdong Province, China) and has a typical subtropical ocean climate with a mean annual SST of 24 °C [23]. Therefore, we hypothesized that the coral microbiome assemblages among different coral species between these two thermal regime regions would be different. To test this, the dominant coral species *Acropora* sp. (scleractinian) and *Porites lutea* (scleractinian), as well as the four common coral species *Galaxea fascicularis* (scleractinian), *Platygyra lamellina* (scleractinian), *Favosites speciosa* (scleractinian) and *Sarcophyton glaucum* (alcyonacea), were sampled from different reef sites of the two reefs. The coral microbiome assemblages were compared via pyrosequencing of 16S rRNA gene and multivariate analysis. 

## 2. Materials and Methods

### 2.1. Sampling Regions and Sample Collection

Coral samples of *Acropora* sp., *G. fascicularis*, *P. lamellina*, *P. lutea*, *F. speciosa* and *S. glaucum* were collected from tropical Sanya Bay reef (SY) (E109.430°–109.490°, N18.210°–18.221°) and subtropical Daya Bay reef (DY) (114.411°–114.650°E, 22.841°–22.780°N) in the Northern SCS in August 2018 (Table 1). At each sampling reef, coral colonies from at least four reef sites were collected (distance > 100 m). Coral fragments of approximately 1 cm^2^ were picked and washed with 0.22 μm polycarbonate membrane filtered seawater at least three times. The corals were identified based on morphology and molecular analysis of the *cytb* gene (Additional file 1). 

To identify bacterial community composition in the seawater, around 0.5 L of seawater samples from at least four sites in each reef were collected and filtered using 0.22-μm polycarbonate membranes (Whatman GF/F, 47 mm). Then, the filters from each reef region were mixed separately. All samples were preserved in RNAlater^®^ at 4 °C in the field and stored at −80 °C in the laboratory until DNA extraction.

### 2.2. Environmental Data Collection

The mean annual SST heatmap was generated by ODV 5.1.7 software (https://odv.awi.de/) using the data retrieved from the NOAA (https://coralreefwatch.noaa.gov/product/5km/index_5km_composite.php). During sampling periods, the SST, salinity and depth of tropical and subtropical reef regions (at least four reef sites in each reef) were measured by a multi-parameter conductivity, temperature and depth profiling unit-CTD (Idronaut, Italy). Dissolved oxygen (DO) was measured using a YSI 6600V2-02 multi-parameter instrument (Xylem-YSI, Yellow Springs, OH, USA). Inorganic nutrients (nitrite-NO_2_^-^, nitrate-NO_3_^-^, ammonia-NH_4_^+^ and phosphate-PO_4_^3-^) were analyzed using a Lachat QC8500 Flow Injection Autoanalyzer (Lachat Instruments, Loveland, CO, USA) (Additional file 2: Appendix A).

### 2.3. DNA Extraction, Amplification, Pyrosequencing and Data Processing

Total genomic DNA of coral and seawater samples were extracted using a PowerSoil DNA Isolation Kit (MoBio Laboratories, Carlsbad, CA, USA) according to the manufacturer’s protocol. Because V3–V4 has been the preferred region for next-generation studies and has been widely used for analysis biodiversity of marine bacteria and marine symbiotic bacteria [24,25], we selected V3 and V4 hypervariable regions for coral microbiome analysis. The V3 and V4 hypervariable regions of bacterial 16S rRNA genes were amplified with the primers 341F (5′-CCTAYGGGRBGCASCAG-3′) and 806R (5′-GGACTACNNGGGTATCTAAT-3′).

For pooling multiple samples in one run of Illumina sequencing (MiSeq), a unique 12-mer tag for each DNA sample was added to the 5′ end of both primers. Each sample was PCR-amplified in a 50 μL reaction, which contained 25 μL Multiplex Taq (Qiagen, Hilden, Germany), 10 mM of each primer, 60 ng of genomic DNA, and DNase-free water to make a total volume of 50 μL. Cycling conditions were as follows: 94 °C for 5 min followed by 30 cycles of denaturation at 94 °C for 30 s, annealing at 52 °C for 30 s, extension at 72 °C for 30 s, and a final extension at 72 °C for 10 min. The PCR products were validated by Agilent 2100 Bioanalyzer (Agilent Technologies, Palo Alto, CA, USA), and quantified by Qubit^®^ 3.0 Fluorometer (Life Technologies, New York, NY, USA). Finally, the PCR products were performed using a 2 × 300 paired-end (PE) configuration. Base calling was done by the MiSeq Control Software (MCS) embedded in the MiSeq instrument. 

Raw reads (SUB6472444) of the 16S rRNA gene sequences were processed using the QIIME1 (quantitative insights into microbial ecology) platform [26]. In brief, primer sequences were trimmed, paired-end reads merged, and QIIME’s default quality-control parameters were used for quality control. Chimeric sequences were removed using usearch 6.1 [27] implemented in QIIME. After quality control, a total of 4,798,543 (average 65,733 reads per coral sample) 16S rRNA gene sequences were retained, which were clustered into 7604 operational taxonomic units (OTUs) at a 97% similarity level using QIIME’s subsampled open reference OUT-picking protocol [26] (Additional file 2: Appendix A). OTUs containing only three 16S rRNA sequences across all samples were removed from the generated OTU table. The sequences coverage (goods_coverage) of each coral sample was close to 100% (Additional file 2: Appendix A). The representative sequence of each OTU was assigned to different taxonomic levels by the UCLUST consensus taxonomy assigner, and sequences were aligned against the Greengenes database (gg_13_8) using PyNAST [28]. 

The core microbiome is ecologically and functionally important for the coral holobiont fitness [9]. The definition of the core microbiome in corals is varied among authors, ranging from 30% to 100% of shared bacterial phylotypes among different samples [9]. Here, we defined the coral core microbiome as bacterial phylotypes presented in at least 90% of coral samples.

### 2.4. Network Analysis and Functional Profiling of Coral Microbiome

In each studied reef (tropical and subtropical), we selected the top 600 most abundant OTUs, which accounted for nearly 90% of 16S rRNA sequences in all samples, to perform network analyses using a molecular ecological network (MEN) method [29]. The MENs construction was conducted in MENAP (http://ieg2.ou.edu/MENA) [29]. The relative abundance of OTUs were log-transformed and missing values were filled with 0.01 if paired valid values were available. Similarity matrices (adjacency matrix) were created for each network based on the pairwise Pearson correlation coefficient across the time-series (two-time points lag). The threshold of pairwise Pearson correlation coefficient values between OTUs was identified by a random matrix theory (RMT)-based approach that observed a transition point of nearest-neighbor spacing distribution of eigenvalues from Gaussian to Poisson distribution [30]. Only OTUs found at least in one third of the samples in tropical and subtropical reefs were selected for network construction, respectively. Then the same cut-off of 0.64 was used to construct the microbial community networks. Additionally, ecological networks predicted by R^2^ (R^2^ > 0.8) generated based on the random matrix theory (RMT) should be scale-free [30]. Once the MEN was determined, the topological indices were calculated based on the adjacency matrix. Module detection of each network was based on fast greedy modularity optimization [31]. The most abundant bacteria (top 60) detected in coral samples were visualized by heatmap using package ggplot2 in R software [32].

We applied the Phylogenetic Investigation of Communities by Reconstruction of Unobserved States (PICRUSt2) to predict metagenomic functional content from the 16S rRNA marker genes [33]. The “picrust2_pipeline.py” with default sets was applied to predict metagenomic functional content of coral microbiome based on the 16S rRNA marker genes. The output pathway abundance (MetaCyc metabolic pathway) table of metagenome predictions was further analyzed using STAMP (Statistical Analysis of Metagenomic Profiles) software [34] to identify significantly different functional pathways of coral microbiome between tropical and subtropical reef regions. 

### 2.5. Multivariate Statistics Analyses

A non-metric multidimensional scaling (NMDS) analysis and a permutational multivariate analysis of variance using distance matrices (PERMANOVA) were performed to test the significant differences of coral microbiome among different coral species and between different groups using the vegan package [35] in the R software environment [32]. To test the correlation of environmental factors and hosts with coral microbiome, a RDA (Redundancy analysis) analysis using permutation tests was also conducted in the R software environment [32]. To assess the significant differences of environmental characteristics between tropical and subtropical reefs, we conducted analyses of variance tests (Tukey’s HSD) using the stats package in R software [32]. To determine significant differences between bacterial communities between tropical and subtropical reef regions, as well as between coral and seawater groups, the STAMP software package was used [34]. *P* values were calculated using a two-sided ANOVA similar statistic test. Significant differences of functional pathways of coral microbiome between tropical and subtropical reef regions were also analyzed by STAMP software [34]. 

### 2.6. Availability of Data and Materials

The raw sequence information in this paper has been deposited in the GenBank Sequence Read Archive with accession number SUB6472444.

## 3. Results 

### 3.1. Environmental Parameters at the Study Regions

The mean annual SST of SY (tropical) reef was about 27 °C, which was 3 °C higher than that of the DY (subtropical) reef (Figure 1). During sampling periods, the tropical reef had significantly higher mean values of SST, NO_3_^-^, NH_4_^+^ and PO_4_^3-^ than the subtropical reef (Tukey’s HSD, *p* < 0.001) (Additional file 3: Appendix A). 

### 3.2. Coral Microbiomes and Their Relationships with the Host and Environment

Ordination and molecular ecological network methods were used to investigate the relationships of coral microbiomes with hosts and environmental parameters. The NMDS (Figure 2) showed that the coral microbiome assemblages were grouped into different clusters according to reefs and coral species. The compositions of coral microbiome in the tropical and subtropical reefs were significantly different (PERMANOVA analysis: *R* = 0.3231, *p* < 0.001). The microbiome compositions of same coral species were significantly different between tropical and subtropical reefs, except for *P. lutea* (Additional file 2: Appendix A). In the tropical reef, coral microbiome differed significantly among all coral species, while in subtropical reef, only several coral microbes had significantly different compositions (Additional file 2: Appendix A). It was noted that the coral microbiomes were distinct from bacterial composition in the seawater samples (PERMANOVA analysis: R = 0.597, *P* = 0.015) (Additional file 2: Appendix A). 

The molecular ecological network analysis of coral microbiomes identified 101 and 184 nodes and 274 and 1421 links from the tropical and subtropical networks, respectively (Table 2). Several network topological indices, including average connectivity (avgK), average clustering coefficient (avgCC) and average geodesic distance (avgGD), were greater in the subtropical network (Table 2). These results indicated a more compact ecological network of coral microbiomes in the subtropical corals.

RDA showed that the distribution of major coral microbiome bacterial OTUs was majorly explained by environmental factors (Figure 3). The four environmental variables (SST and NO_3_^-^, NH_4_^+^, PO_4_^3-^) together explained 49.4% of the variance in coral microbiome, while host taxa only explained 11.5% of the variance, indicating that environmental variables served as main factors affecting coral microbiome. Specifically, the coral microbiomes were significantly affected by the SST in tropical and subtropical reefs (permutation tests, *p* < 0.003). Coral microbiomes were mainly clustered into two major groups. One group was formed by microbiomes from the tropical reef and was positively correlated with SST and NO_3_^-^, NH_4_^+^, PO_4_^3-^, and the other group composed by microbiomes from the subtropical reef was negatively correlated with SST and NO_3_^-^, NH_4_^+^, PO_4_^3-^. 

### 3.3. Diversity and Functional Groups of Coral Microbiome 

In total, 56 bacterial phyla were detected in the coral microbiome (Figure 4: Additional file 4). γ- (32%) and α-proteobacteria (19%), Bacteroidetes (14%), Firmicutes (14%), Actinobacteria (6%) and Cyanobacteria (2%) dominated the coral microbiome. At the family level (Additional file 3: Appendix A), Endozoicimonaceae (γ-proteobacteria, 10%), Moraxellaceae (γ-proteobacteria, 9%), Carnobacteriaceae (Firmicutes, 6%), Flavobacteriaceae (Bacteriodetes, 5%), Rhodobacteraceae (α-proteobacteria, 5%), Bifidobacteriaceae (Actinobacteria, 3%), Rhodospirillaceae (α-proteobacteria, 2%), Pelagibacteraceae (α-proteobacteria, 2%), Planococcaceae (Firmicutes, 2%) dominated the coral microbiome. 

As shown in Figure 5, the most abundant OTUs affiliated to different bacterial phylotypes were further designated as core, host specific and environment-related coral microbiome according to their distribution in the coral microbiomes. The OTUs affiliated with *Endozoicomonas* sp., *Psychrobacter pacificensis*, *Granulicatella adiacens*, *Ruegeria lacuscaerulensis*, *Bifidobacterium longum*, *Planococcus maritimus*, Pelagibacteraceae, *Staphylococcus epidermidis*, Piscirickettsiaceae, *Sulfitobacter* sp., *Dietzia natronolimnaea*, *Octadecabacter ascidiaceicola*, *Bacteroides acidifaciens*, *Bradyrhizobium elkani*, *Ulvibacter* sp., *Lactococcus piscium*, *Sphingomonas asaccharolytica*, *Phyllobacterium* sp., *Acinetobacter johnsonii*, Actinomycetales, Methylophilaceae, *Candidatus* Aquiluna rubra and one unclassified bacterium, were coral core microbiome (Figure 5, red triangles). These bacteria were detected in at least 90% of coral samples.

Some OTUs were enriched in the coral samples and were not detected in seawater samples, such as those affiliated to bacterial phylotypes Flavobacteriaceae, Helicobacteraceae, Myxococcales, *Thalassomonas* sp., *Muricauda lutaonensis* and Rhodospirillaceae (Figure 5, red stars), which suggests these bacterial OTUs might be strictly symbiotic bacteria. It was noted that several OTUs affiliated to Flavobacteriaceae, Spirochaetes, Helicobacteraceae were enriched in *Acropora* sp., *S. glaucum* and *P. lutea*, respectively. However, most OTUs affiliated to different bacterial phylotypes were detected in nearly all or several coral species, revealed that host specificity was rare among individual coral species. 

For environment-related coral microbiome, a total of 13 OTUs affiliated to different bacterial phylotypes were enriched in tropical reef (Figure 5, red cycles) and 41 OTUs affiliated to different bacterial phylotypeswere enriched in subtropical reef (Figure 5, blue cycles). Several OTUs affiliated to *Ruegeria lacuscaerulensis*, *Bradyrhizobium elkanii*, *Sphingomonas asaccharolytica* etc., remained with a stable abundance among different coral samples between the tropical and subtropical reefs. Specifically, two thermophilic marine bacteria, *Muricauda lutaonensis* and *Truepera* sp., were enriched in tropical corals. 

### 3.4. Microbial Functional Profiles Change Between the Tropical and Subtropical Reefs

A total of 38 functional pathways related to coral microbiome were significantly different abundance between the tropical and subtropical reefs (Table 3). Notably, pathways related to the synthesis of vitamin B1, vitamin B6, biotin, folate, heme, NAD, coenzyme and isopentenyl were significantly enriched in the tropical reef, whereas the vitamin K2 biosynthetic pathway was enriched in the subtropical reef. Additionally, we observed that some pathways related to fatty acid elongation (saturated), chlorophyll a biosynthesis and L-methionine biosynthesis by sulfhydrylation were enriched in the subtropical reef. Other enriched pathways, including threonine and ornithine degradation, aspartate biosynthesis, aromatic compounds degradation and man-made Nylon-6 oligomer degradation, were also predicted. 

## 4. Discussion

In this study, the coral microbiome assemblages among 73 samples across six typical coral species living in two naturally thermal regimes (tropical and subtropical) of the SCS were explored using pyrosequencing of bacterial 16S rRNA gene and multivariate analysis. Our present results showed that coral microbiome assemblages and underlying functional profiles were different between the tropical and subtropical reefs, and the differences in coral microbiome assemblages were more strongly determined by environmental factors (e.g., SST) than host specificity. Most importantly, we found that besides bacteria inferred to play potential roles in host nutrients metabolism, several keystone bacteria which were previously detected as human microbiome, plant rhizobium, as well as intracellular symbiont of amoebae and other marine hosts, were retrieved in nearly all explored corals in the SCS. 

### 4.1. Effects of Hosts and Thermal Regimes on Coral Microbiome Assemblages

The present study revealed that the coral microbiome assemblage composition was significantly different from bacterial community composition in seawater, suggesting the specific control of the coral microbiome by hosts. This was in agreement with most of the studies published to date [10,13,25]. In a thermal regime where coral species have similar environmental conditions, we observed that the top OTUs were shared by all or several coral species, suggesting rare host specificity among individual coral species. The possible explanation for this result is that rare host specificity may limit the risk of extinction and provide more immediate fitness benefits to holobiont, such that selection may favor evolution toward a generalist strategy [36].

On the other hand, we observed that coral microbiome assemblage composition varied across habitats. This observation was consistent with previous studies performed by Ziegler et al. [37] and Brener-Raffalli et al. [19]. For instance, Ziegler et al. [37] reported that *Acropora hyacinthus* in the back reef pools of Ofu Island in different thermal habitats had distinct microbiome assemblages. They suggested that temperature might be a key factor determining coral microbiome assemblage. In our study, one exception was *P. lutea* which exhibited relative stable microbiome composition between tropical and subtropical reefs (*p* = 0.142), revealing inter-species differences as well. In the Indo-Pacific Ocean, previous studies have implied that *Porites*-algae symbiotic associations were stable over broad geographical scales and temperature ranges, and *Porites* holobiont was persistent to elevated SST [38,39,40]. Taken together, those results suggested that *Porites* may have more conserved microbiome than other coral species. 

Furthermore, the present study revealed that coral microbiome in tropical reef exhibited more heterogeneity than that in subtropical reef, and the tropical reef microbiome had a more un-compacted ecological network than subtropical reef microbiome. It is commonly believed that the optimum temperature for coral growth and development ranged from 23 °C to 27 °C. In this study, the mean SST of the tropical sampling sites was about 31 °C (Additional file 3: Appendix A), which is a thermal stress for coral as it is above the optimum temperature and near to the threshold temperature of coral bleaching [41]. According to the Anna Karenina principle (AKP) [42], environmental stresses can lead to transitions of microbiome networks from a stable to unstable state. This may explain why the ecology network of tropical reef microbiome is more heterogenous and un-compacted. Overall, the present results indicated that the coral microbiome assemblages in most explored corals were more strongly determined by environmental factors (e.g., SST) than host specificity, but inter-species difference also existed. 

### 4.2. Major Members and Potential Roles of Coral Microbiome

The potential roles of most abundant bacterial phylotypes detected in coral holobiont were listed in Figure 6. These listed bacteria might be involved in several important biological and ecological processes in coral holobiont, e.g., carbon, nitrogen, sulfur cycles and detoxification, regulatory and/or defense roles, environmental stress. A previous study indicated that carbon dioxide was mainly fixed into carbohydrates by Symbiodiniaceae in coral holobiont [43]. Our present results revealed that *Synechococcus* sp., the most abundant photosynthetic fixer in the global oceans [44], was detected in most explored corals. The grazing of *Synechococcus* by Symbiodiniaceae was reported in a previous study [45]. Thus, we proposed that *Synechococcus* might also supply carbon sources to coral or Symbiodiniaceae. The organic carbon (e.g., mucus of coral, the cell wall of Symbiodiniaceae) in coral holobiont could be utilized by *Ulvibacter* which has been reported as a robust polysaccharide utilizer in seaweeds [46]. The nitrogen source of coral holobiont might come from *Dietzia*, *Bradyrhizobium* and *Phyllobacterium* which are widely reported as nitrogen-fixing bacteria in the plant rhizobium [47]. In addition, potential denitrifying (e.g., *Nitratireductor*), nitrite-oxidizing (e.g., *Nitrospira*) and ammonia-oxidizing (e.g., *Nitrosomona*) bacteria were also detected in coral holobiont. Together, those bacteria might be involved in the complete nitrogen cycle in coral holobiont [48]. The *Ruegeria*, *Sulfitobacter* and *Prosthecochloris* can participate in the sulfur metabolism [49]. Hydrogen sulfide is toxic to a wide range of eukaryotic organisms and might lead to the initiation of coral blank band disease. *Sulfitobacter* and *Prosthecochloris*, serving as potential sulfur-oxidizing bacteria, might oxidize holobiont-accumulated hydrogen sulfide to sulfate, thus contributing to coral health through detoxification of reduced sulfur compounds [50]. DMSP (dimethylsulfoniopropionate) and DMS (dimethylsulfide) are important compounds in the global sulfur cycle. Previous study has indicated that the DMSP in coral holobiont is produced by both coral and Symbiodiniaceae [51]. The generated DMSP might be degraded into climate-active gas DMS via the bacterial cleavage pathway by bacteria detected in corals, such as *Ruegeria*, *Roseovarius* and *Psychrobacter* [51]. The variation of activity and abundance of those bacteria would be important for regulating nutrient metabolisms in coral holobiont.

Interestingly, a number of bacteria, which has been previously detected in human microbiome, e.g., *Bifidobacterium longum*, *Bacteroides acidifaciens*, *Lactococcus piscium*, *Parabacteroides* sp., *Akkermansia muciniphila*, *Muribaculum intestinale*, *Alistipes finegoldii*, *Megasphaera* sp., *Dermabacter vaginalis*, *Gardnerella vaginalis*, *Atopobium vaginae* and *Granulicatella adiacens* [52,53,54], were found in nearly all explored corals. The functions of these bacteria have been widely reported in humans, e.g., a regulatory role in colon walls, a defense barrier enhancer, an intestinal motility modulator, an anti-inflammatory action modulator [55], but their function in the coral holobiont remains unclear. Meanwhile, the bacteria of “*Candidatus* Amoebophilus asiaticus”, *Octadecabacter ascidiaceicola* and *Gemella* sp. were found in several explored corals in this study. “*Candidatus* Amoebophilus asiaticus” was recognized as an intracellular symbiont of amoebae [56], and has been reported to be associated with the tissues of Caribbean corals [57]. The *Octadecabacter ascidiaceicola* was previously reported in sea squirt [58]. The *Gemella* sp. was only detected in commensals of the mucous membranes of humans and some other warm-blooded animals [59]. These bacteria added a further degree of intricacy to coral holobiont symbioses.

A previous study has suggested that coral’s acclimation/tolerance to thermal or other environmental stresses is related to both host and their hosted Symbiodiniaceae [60]. Recently, the study conducted by Ziegler et al. [37] firstly linked bacteria (e.g., *Inquilinus*, a thermal tolerant bacterial genus) with coral heat tolerance. In the present study, we found that *Muricauda lutaonensis* (a thermophilic bacterium isolated from coastal hot spring) [61] and *Truepera* sp. (a bacterium with high thermal and radiation resistant) [62] were significantly enriched in tropical SY thermal regime. The microbial functional profiles further revealed that pathways related to vitamins and coenzymes synthesis were significantly enriched in the SY thermal regime. The vitamins and coenzymes have been shown to be related to growth, symbiosis and various stresses of organisms [63]. Based on these results, we proposed that the above differences would be related to the thermal adaptation of coral holobiont under elevated temperature. In addition, the present results showed that short chain fatty acids (e.g., acetate/lactate and butanoate) related pathways were enriched in coral microbiome. Short chain fatty acids, the end products of fermentation of dietary fibers by the anaerobic gut microbes, have been shown to exert multiple beneficial effects on mammalian energy metabolism and host-microbes interactions [64]. In coral holobiont, the fibers are mainly produced by Symbiodiniaceae. Thus, a possible scenario was that bacteria (e.g., *Bifidobacterium longum*) associated with coral could utilize fibers of Symbiodiniaceae for producing short chain fatty acids, which may affect “bacteria-algae-host” interactions.

Except for *Actinomycelates* and *Endozoicomonas,* most aforementioned bacteria have not been reported in previous studies [9,65,66], and the inconsistent results would be aroused by study design, target habitat, sample size, sequencing approach and the analysis tools used for analysis coral microbiome [65].

## 5. Conclusions

This study disentangled the multiple relationships of the coral microbiome with both host taxa and thermal regime habitants. The results indicated that the coral microbiome assemblages in most explored corals were more strongly determined by environmental factors (e.g., SST) than host specificity, but inter-species difference also existed in coral species. Meanwhile, this study revealed that the high SST and nutrients in the tropical SY reef resulted in a more heterogeneous community and an un-compact ecological network related to coral microbiome, implied that the community structure of coral microbiome will be changed from a compacted to an un-compacted network under environmental stresses. Interestingly, besides bacteria inferred to play potential roles in nutrients metabolisms, several keystone bacteria in human microbiome, plant rhizospheric microbiome and two intracellular symbionts of marine hosts were detected in corals in the present study. Meanwhile, several thermophilic bacteria (e.g., *Muricauda lutaonensis* and *Truepera* sp.) were detected in explored corals from tropical SY reef. Most of those bacteria have not been reported in previous studies, which provides important clues and opens a path for further targeted studies of coral holobiont symbiosis under future global climate change and anthropogenic disturbance.

## Figures and Tables

**Figure 1 microorganisms-08-00604-f001:**
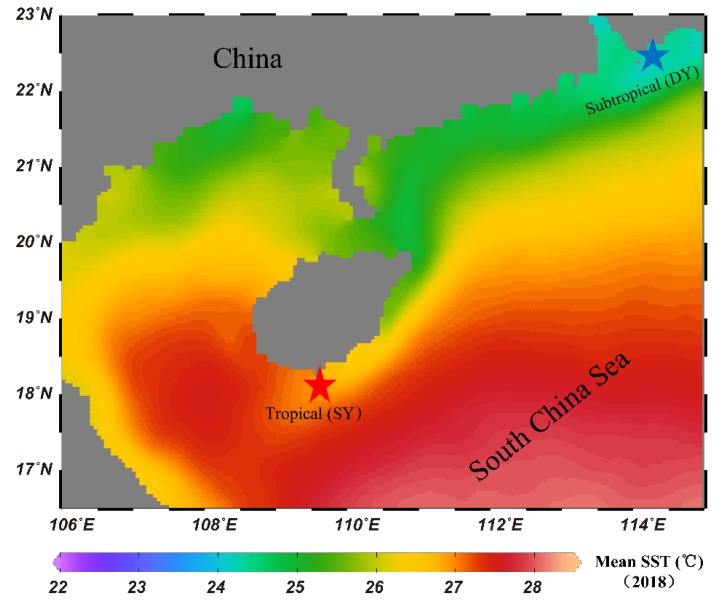
Sampling maps: Red star (★) represents the tropical reef (SY), Hainan Island, in the northern part of the SCS; Blue star (★) represents the subtropical reef (DY), which is located southeast of Shenzhen City of Guangdong Province, China. The mean annual SST heatmap was generated using data from the NOAA (https://coralreefwatch.noaa.gov/product/5km/index_5km_composite.php) for 2018.

**Figure 2 microorganisms-08-00604-f002:**
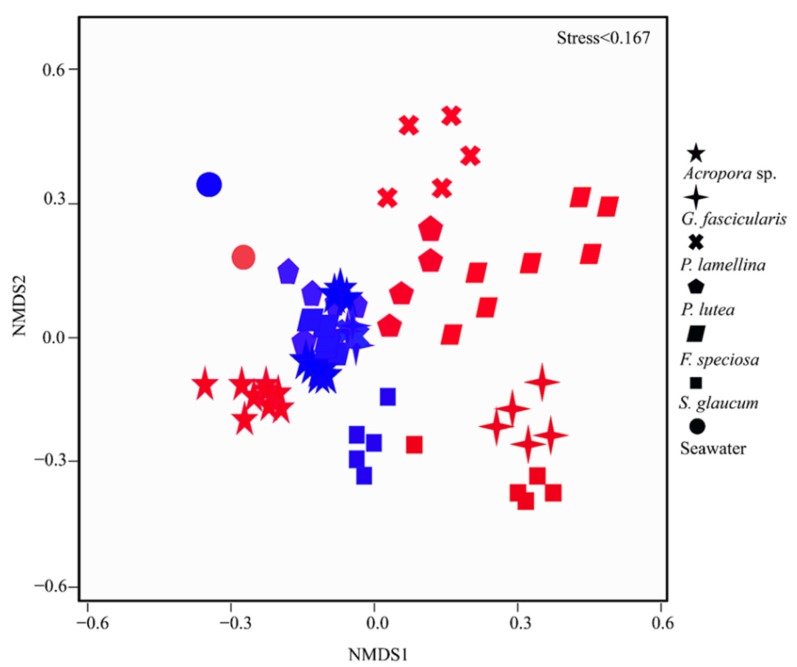
Non-metric multidimensional scaling (NMDS) ordination of coral microbiome assemblages based on Bray–Curtis distance. Different symbols represent different coral species (the names of different coral species and corresponding symbols were illustrated on the right of the NMDS ordination plot). The solid cycle represents seawater. Red symbols indicate samples from the SY reef, and blue symbols represent samples from the DY reef.

**Figure 3 microorganisms-08-00604-f003:**
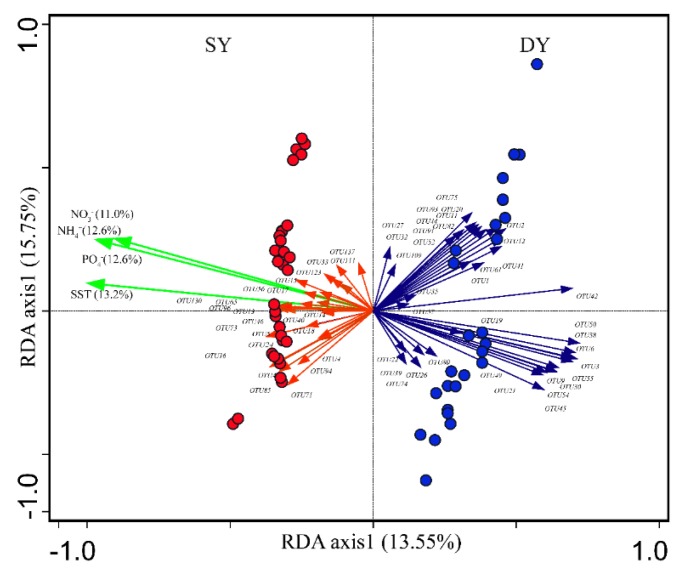
RDA (Redundancy analysis) depicting the relationships of coral microbiomes (top 60 OTUs) with hosts and environmental parameters. The arrows represent coral microbiome enriched in the tropical SY (red) and the subtropical DY (blue) reefs, respectively. The green arrows represent environmental factors. The solid cycles represent coral samples from the tropical SY (red) and the subtropical DY (blue) reefs, respectively.

**Figure 4 microorganisms-08-00604-f004:**
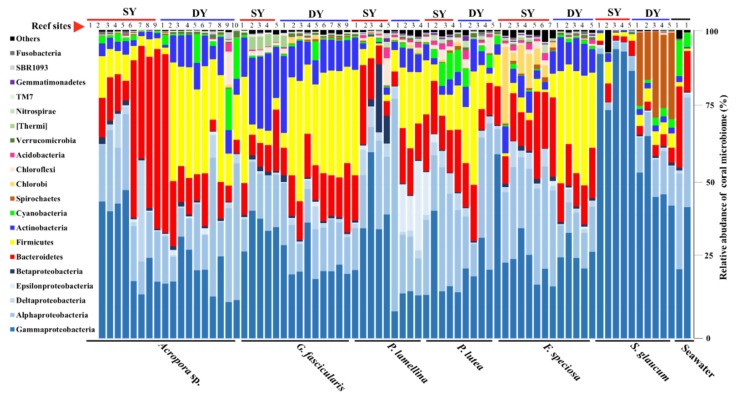
Bar plot of the relative abundance of coral microbiome at the phylum level. Each bar represents the relative abundance of different bacterial phylum in coral microbiomes or seawater samples.

**Figure 5 microorganisms-08-00604-f005:**
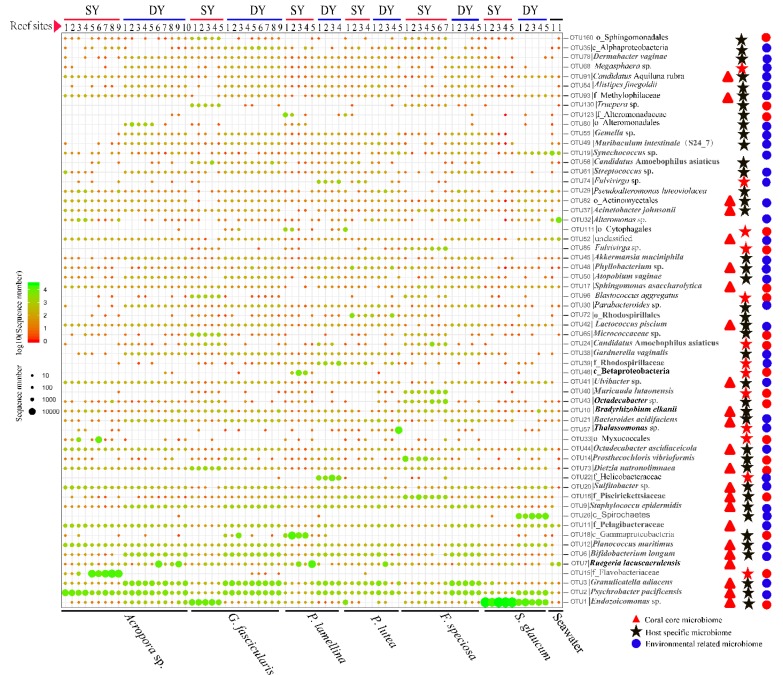
Profiles of the top 60 abundant OTUs affiliated to different bacterial phylotypes detected in explored coral samples. Relative abundance was log10 transformed for plotting. The bottom panel represents coral species. The top panel represents reef sites of each coral species from the tropical SY and subtropical DY reefs. The red triangle of the right panel represents coral core microbiome. The stars represent the host-specific microbiome, and the solid cycles represent environmental related microbiome (red cycles represent coral associated bacteria enriched in the tropical SY reef; blue cycles represent coral associated bacteria enriched in the subtropical DY reef).

**Figure 6 microorganisms-08-00604-f006:**
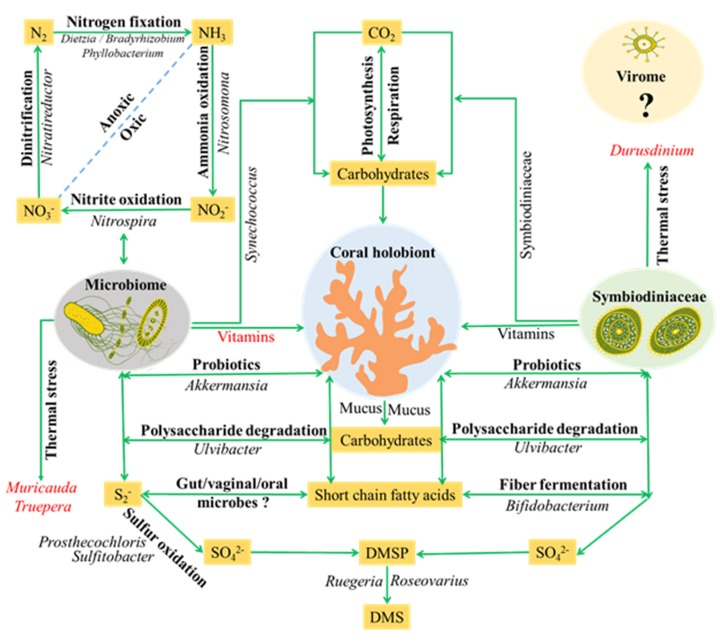
A schematic representation illustrating potential roles and relationships among coral, micriobiome and Symbiodianecean. Potential roles might be played in host nutrients metabolism (carbon, nitrogen and sulfur cycles) and thermal stress. The bacteria genus denoted in italics were mapped onto the well-known ecological processes. The gut/vaginal/oral microbes and Virome are likely to be important targets in future studies. The question mark (?) means that the roles of microorganisms in coral holobiont are unknow at present. The blue dot line represents two kinds of notrogen cycles (Oxic and Anoxic).

**Table 1 microorganisms-08-00604-t001:** Coral samples from tropical and subtropical reefs of the South China Sea (SCS). Including sampling regions (latitudes and longitudes), climate, coral samples information.

Regions	Climate	Coral	Reef Sites (*n*) ^a^	Date
Sanya Bay (SY) E109.430°–109.490°, N18.210°–18.221°	Tropical	*A.* sp.^b^	9	2018-08-25
		*G. fascicularis*	5	2018-08-25
		*P. lamellina*	5	2018-08-25
		*P. lutea*	4	2018-08-26
		*F. speciosa*	7	2018-08-26
		*S. glaucum*	5	2018-08-25
Daya Bay (DY) E114.411°–114.650°, N22.841°–22.780°	Subtropical	*A.* sp.	10	2018-08-27
		*G. fascicularis*	9	2018-08-27
		*P. lamellina*	4	2018-08-27
		*P. lutea*	5	2018-08-27
		*F. speciosa*	5	2018-08-27
		*S. glaucum*	5	2018-08-27

^a^ The number of coral sites for sampling. ^b^
*Acropora* sp.

**Table 2 microorganisms-08-00604-t002:** Major properties of molecular ecological network.

	Nodes	Links	R^2^ of Power-Law	AvgK ^a^	AvgCC	AvgGD
Tropical (SY)	101	274	0.833	5.426	0.128	3.148
Subtropical (DY)	184	1421	0.800	15.446	0.547	4.125

^a^ avgK: average connectivity; avgCC, average clustering coefficient; avgGD, average geodesic distance.

**Table 3 microorganisms-08-00604-t003:** Significantly different functional pathways related to coral microbiome between tropical (SY) and subtropical (DY) reefs (The↑and↓represent up- and down-regulated pathways, respectively).

PathwayID	Pathway Description	Variation Trends (SY vs DY)	*P* Values
PWY-5896	Vitamin K2 synthesis	↑	6.84 × 10^−5^
PWY-6897	Vitamin B1 synthesis	↓	3.32 × 10^−4^
PYRIDOXSYN-PWY	Vitamin B6 synthesis	↓	1.21 × 10^−7^
PWY-6519	Biotin synthesis	↓	3.32 × 10^−4^
FOLSYN-PWY	Folate synthesis	↓	1.28 × 10^−5^
HEMESYN2-PWY	Heme b biosynthesis by anaerobic	↓	3.32 × 10^−4^
PWY-5920	Heme b biosynthesis from glycine	↓	1.89 × 10^−10^
NAD-BIOSYNTHESIS-II	NAD biosynthesis	↓	1.36 × 10^−8^
POLYISOPRENSYN-PWY	Isopenteny biosynthesis (IPP)	↓	1.94 × 10^−7^
PANTO-PWY	Phosphopantothenate biosynthesis I (coenzyme)	↓	1.90 × 10^−11^
PWY-5198	Factor 420 biosynthesis (coenzyme)	↓	4.45 × 10^−7^
P23-PWY	CO_2_ fixation by reductive TCA	↓	3.24 × 10^−8^
PWY-7024	CO_2_ fixation by 3-hydroxypropanoate cycle	↑	8.20 × 10^−6^
PENTOSE-P-PWY	Pentose phosphate pathway (PPP)	↓	2.12 × 10^−6^
PWY-5659	GDP-mannose biosynthesis (LPS)	↑	1.13 × 10^−5^
GALACTARDEG-PWY	D-galactarate (sugar acid) degradation	↓	9.68 × 10^−5^
PWY-5100	Pyruvate fermentation to acetate and lactate	↓	5.19 × 10^−4^
GOLPDLCAT-PWY	Glycerol degradation: propanediol	↑	5.40 × 10^−5^
PWY-5677	Succinate fermentation to butanoate	↓	6.37 × 10^−8^
P122-PWY	Heterolactic fermentation: lactate/CO_2_/CH_3_OH	↑	3.37 × 10^−5^
FASYN-ELONG-PWY	Fatty acid elongation-saturated	↑	2.60 × 10^−6^
PWY-5971	Fatty acid biosynthesis: palmitate	↓	2.50 × 10^−7^
PWY-5529	Bacteriochlorophyll a biosynthesis	↓	1.21 × 10^−8^
PWY-5531	Chlorophyll a biosynthesis	↑	3.24 × 10^−4^
PWY-6545	Pyrimidine biosynthesis	↑	6.12 × 10^−4^
THREOCAT-PWY	L-threonine degradation	↑	4.33 × 10^−7^
ORNDEG-PWY	L-ornithine degradation (putrescine biosynthesis)	↓	1.05 × 10^−6^
PWY-5347	L-methionine biosynthesis (transsulfuration)	↓	4.89 × 10^−7^
PWY-5345	L-methionine biosynthesis (by sulfhydrylation)	↑	4.86 × 10^−5^
HOMOSER-METSYN-PWY	L-methionine biosynthesis I	↓	3.46 × 10^−4^
PWY-5028	L-histidine degradation II	↓	3.46 × 10^−4^
ASPASN-PWY	L-aspartate and L-asparagine biosynthesis	↓	3.28 × 10^−8^
PWY0-781	L-aspartate biosynthesis	↓	3.29 × 10^−8^
PWY0-1338	Polymyxin resistance (antibiotic resistance)	↑	3.24 × 10^−4^
PWY-6071	Aromatic compound degradation: phenylethylamine	↓	5.15 × 10^−7^
PWY-6210	Aromatic compound degradation:2-aminophenol degradation	↓	1.36 × 10^−4^
PWY-5419	Aromatic compound degradation: catechol	↑	7.74 × 10^−6^
P621-PWY	Nylon-6 oligomer degradation	↓	3.76 × 10^−6^

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
