# Peer review of "Unraveling Heterogeneity of Coral Microbiome Assemblages in Tropical and Subtropical Corals in the South China Sea"

_microorganisms, 2020, doi:10.3390/microorganisms8040604_

Round 1
Reviewer 1 Report
This is an interesting, well-planned study. The main finding reported in this manuscript is that coral microbiome dysbiosis across host taxa and thermal regime habitatants. Moreover, this study reported that the high surface seawater temperature and nutrients associated with more bacterial diversity in tropical coral microbiome and had a high un compacted ecological network due to environmental stress. Study identify important thermophilic bacteria such as Muricauda lutaonensis and Truepera sp. from corals of SY reef.It has been written in a clear and comprehensive way and reflects the purpose of the study. However, more robust microbiome analysis is needed that may improve the quality of the manuscript.
Comment
1. Additional files and supplementary figures are missing in the submitted manuscript. Can author provide the same?
2. In the Method section, author should mention why V3 and V4 hypervariable region were selected for 16S microbiome analysis. Does author optimize amplification cycle of first PCR as 30 cycle might enriched dominant bacteria and can give biased result.
3. It will be good if author can provide raw data of microbiome in the supplementary material.
4. For reader convenience, Author should provide quality profile of forward reads (R1) and corresponding reverse reads (R2) of the presentative same samples. The analysis can be performed using a DADA2 pipeline.
5. For reader convenience, can author provide alpha diversity measures (Shannon Diversity) of coral species and different reef region.
6. Author should consider adding a figure showing the relative abundance of important thermophilic and other bacteria at genus label in different coral species and different reef region.
7. Some of the symbol is missing in the brackets of figure legend 2
8. Manuscript needs to be proofread for proper English grammar and small mistakes -a careful read through to correct these errors will greatly increase readability.
Author Response
Corrections to the manuscript
Dear Editor and Reviewers,
We appreciate the time and efforts by the editor and reviewers in reviewing our manuscript entitled “Unraveling heterogeneity of coral microbiome assemblages in tropical and subtropical corals in the South China Sea”(Manuscript ID: microorganisms-760544). The reviewers’ comments and suggestions help improve our manuscript greatly. We have fully considered reviewers’ comments and revised the manuscript accordingly. In the revised manuscript, all corrections have been marked in red. The detailed responses and corrections are listed below.
Reviewer 1:
This is an interesting, well-planned study. The main finding reported in this manuscript is that coral microbiome dysbiosis across host taxa and thermal regime habitatants. Moreover, this study reported that the high surface seawater temperature and nutrients associated with more bacterial diversity in tropical coral microbiome and had a high un compacted ecological network due to environmental stress. Study identify important thermophilic bacteria such as Muricauda lutaonensis and Truepera sp. from corals of SY reef. It has been written in a clear and comprehensive way and reflects the purpose of the study. However, more robust microbiome analysis is needed that may improve the quality of the manuscript.
(1) Additional files and supplementary figures are missing in the submitted manuscript. Can author provide the same?
Response: We have submitted all additional files and supplementary figures together with the manuscript and cover letter to the journal. The reviewer may find the additional files in the journal website.
(2) In the Method section, author should mention why V3 and V4 hypervariable region were selected for 16S microbiome analysis. Does author optimize amplification cycle of first PCR as 30 cycle might enriched dominant bacteria and can give biased result.
Response: We added following sentence to the method section to explain why we selected V3 and V4 regions for this study:
Because V3-V4 has been the preferred region for next-generation studies and has been widely used for analysis biodiversity of marine bacteria and marine symbiotic bacteria (i.e., Dyksma et al., The ISME Journal|(2016)|DOI:10.1038/ismej.2015.257; Ren et al., Applied and Environmental microbiology|(2018)|DOI:10.1128/AEM.02088-18; Cai et al., Scientific Reports| (2018) 8:2428| DOI:10.1038/s41598-018-20515-w), we selected V3 and V4 hypervariable region for coral microbiome analysis.
Although 30 cycles of PCR amplification might enrich dominant bacteria and give biased result, it is commonly used for studying marine bacterial diversity (Cai et al., 2018). In addition, because bacterial concentration in marine water is lower than many other environments, such as soil and waste waters, if we use less PCR cycles we may not get PCR products.
(3). It will be good if author can provide raw data of microbiome in the supplementary material.
Response: The size of raw data set of microbiome is too large to upload as supplementary material, so the raw data set of microbiome has been submitted to public database of NCBI with association number of SUB6472444.
(4). For reader convenience, Author should provide quality profile of forward reads (R1) and corresponding reverse reads (R2) of the presentative same samples. The analysis can be performed using a DADA2 pipeline.
Response: the quality profile (Q20) of R1 and R2 reads the same samples was added in additional file 2-Table S2.
(5). For reader convenience, can author provide alpha diversity measures (Shannon Diversity) of coral species and different reef region.
Response: Shannon Diversity of coral species and two reef regions was calculated and added in additional file 2-Table S2 according to your suggestion.
(6). Author should consider adding a figure showing the relative abundance of important thermophilic and other bacteria at genus label in different coral species and different reef region.
Response: The relative abundance information of the most abundant bacteria (including thermophilic bacteria such as Muricauda lutaonensis ) at OUT/species/genus levels in different coral species and different reef regions can be seen in Figure 5. In addition, more detailed information was supplied in additional file 4.
(7). Some of the symbol is missing in the brackets of figure legend 2
Response: We have changed revised the figure legend 2 as “Figure 2. NMDS ordination of coral microbiome based on Bray-Curtis distance. Different symbols represent different coral species (the names of different coral species and corresponding symbols were illustrated on the right of NMDS ordination plot). The solid cycle represents seawater. Red symbols indicate samples from SY reef, and blue symbols represent samples from DY reef”.
(8). Manuscript needs to be proofread for proper English grammar and small mistakes -a careful read through to correct these errors will greatly increase readability.
Response: We have carefully revised certain mistakes throughout the manuscript, and the newly uploaded manuscript has been revised by a native English speaker.
Thanks for the meticulous checking and helpful suggestions.

Reviewer 2 Report
The paper describes the microbial community of corals in the South China Sea. The experimental outline is well designed. The discussion of the results presents an interesting view of the possible effect of thermal change of marine waters. I also appreciated that the paper does not only describe the microbial strains but also had an insight on the expressed metabolic pathway.
Just a comment. For future comparisons, I would appreciate just few more details about the samples collection as the depth where corals have been picked up.
Author Response
Dear Editor and Reviewers,
We appreciate the time and efforts by the editor and reviewers in reviewing our manuscript entitled “Unraveling heterogeneity of coral microbiome assemblages in tropical and subtropical corals in the South China Sea”(Manuscript ID: microorganisms-760544). The reviewers’ comments and suggestions help improve our manuscript greatly. We have fully considered reviewers’ comments and revised the manuscript accordingly. In the revised manuscript, all corrections have been marked in red. The detailed responses and corrections are listed below.
Reviewer 2:
The paper describes the microbial community of corals in the South China Sea. The experimental outline is well designed. The discussion of the results presents an interesting view of the possible effect of thermal change of marine waters. I also appreciated that the paper does not only describe the microbial strains but also had an insight on the expressed metabolic pathway.
- Just a comment. For future comparisons, I would appreciate just few more
details about the samples collection as the depth where corals have been picked up.
Response: Environmental factors during coral sampling, including SST, Salinity, N/P concentration and Depth can been seen in additional file 3- Figure S1.

Round 2
Reviewer 1 Report
All the concern have been clarified by author and presented in revisied manuscript.